# Immersive Therapy for Improving Anxiety in Health Professionals of a Regional Hospital during the COVID-19 Pandemic: A Quasi-Experimental Pilot Study

**DOI:** 10.3390/ijerph19169793

**Published:** 2022-08-09

**Authors:** Marta Linares-Chamorro, Neus Domènech-Oller, Javier Jerez-Roig, Joel Piqué-Buisan

**Affiliations:** 1Department of Psychology, Fundació Hospital d’Olot i Comarcal de la Garrotxa, 17800 Olot, Spain; 2Department of Knowledge and Innovation, Fundació Hospital d’Olot i Comarcal de la Garrotxa, 17800 Olot, Spain; 3Centre for Health and Social Care Research (CESS), Research Group on Methodology, Methods, Models and Outcomes of Health and Social Sciences (M_3_O), Faculty of Health Sciences and Welfare, University of Vic-Central University of Catalonia (UVic-UCC), 08500 Vic, Spain

**Keywords:** immersive therapy, virtual reality, health professionals, anxiety

## Abstract

Background: Immersive therapy through virtual reality represents a novel strategy used in psychological interventions, but there is still a need to strengthen the evidence on its effects on health professionals’ mental health. Objective: To analyze the results of immersive therapy through virtual reality on the levels of anxiety and well-being of the health professionals working in a regional hospital in Olot (Spain). Methods: Pilot quasi-experimental study including a group of 35 female (mean age = 45.7, SD = 8.43) health professionals who undertook immersive therapy for 8 weeks. The intervention was implemented through virtual reality, and its effect on anxiety levels and well-being was evaluated through the Hamilton and Eudemon scales, respectively. Data on age, gender, active pharmacological or psychological treatment, mental health disorders and number of sessions were also collected. Results: A statistically significant (*p* < 0.001) improvement in anxiety and well-being was found, with large and moderate effect sizes (0.90 and 0.63 respectively). In addition, these changes were clinically significant. No significant associations were found between the improvements and the different variables, but a greater trend was identified among the group of professionals with untreated or unidentified levels of anxiety. Conclusion: This group of health professionals showed a statistically and clinically significant improvement in anxiety and well-being after the application of immersive therapy using virtual reality. Further studies with a control group are necessary to further analyze this novel intervention.

## 1. Introduction

Healthcare professionals perform their routinary work in stressful environments, in continuous contact with people who have problems and who suffer. Constant and uncontrolled stress represents an exogenous factor for the genesis of symptoms of affective disorders related to mood. The COVID-19 pandemic has had a highly negative impact on the emotional health of healthcare professionals [1,2,3,4,5]. The negative effect of the pandemic on the well-being of healthcare workers has been evaluated in different studies [4], mainly analyzing the effect of well-being on mental health [6,7,8,9,10]. According to a recent study, almost half of the healthcare professionals in Spain have a high risk of suffering a mental disorder after the first wave of the COVID-19 pandemic; 45.7% are at high risk for some type of mental disorder, and 3.5% with present active suicidal ideation (death wishes and persistent thoughts of wanting to end one’s own life). Furthermore, 14.5% have a disabling mental disorder, with clear negative repercussions on their professional and social life. By condition, 28.1% present with depression, 22.5% anxiety, almost one in four panic attacks, 22.2% post-traumatic stress disorder and a little more than 6% with substance abuse. In addition, presenting with a mental disorder before the pandemic doubles the risk of suffering from one again as a result of COVID-19 [11].

Regarding the situation of the population studied here, the pandemic has brought to the surface different needs in the emotional management of the professionals working in the Hospital of Olot (currently a staff of approximately 550 workers) [12]. They have been subjected to sustained stress with the risk of generating adaptive problems, changes in behavior, altered mood, somatic or cognitive complications and even severe mental pathology. In this context, it was necessary to reinforce a portfolio of services focused on the emotional wellbeing of the workers, enhancing individual psychological intervention to provide emotional support, advice, guidance and guidelines, i.e., strategies aimed at caring for the emotional wellbeing of professionals. Prior to carrying out this research, 24% of them (120 workers) had requested psychological support from the psychology service of the center. Of these, 25% were discharged after 4 weeks, 48.3% after 5 to 7 weeks, 22.5% after 8 weeks, and 4.2% were still in active psychological treatment when starting this study (April 2021).

Faced with this situation, the research proposed an alternative intervention based on autonomy and using a new immersive reality technology as support tool to address the increase in anxiety among professionals and provide them with another resource to improve emotional well-being. For its implementation, the immersive projection technology MK360 was introduced to transform a room into a multi-sensory interactive experience that can be shared among several people without the use of virtual reality (VR) goggles. This technology covers the user’s field of vision (180° horizontal view × 120° vertical view) with full high definition resolution and reactive audio, and it allows for interaction with the content projected through a smartphone application. It connects to an online platform of immersive content with different programs, including a series of well-being and mindfulness experiences, nature videos and interactive experiences and games that allow the user to control and interact with the elements projected. It enhances the participants’ experience by allowing them to relax and improve their emotional well-being, reduce the perception of pain, generate exercises and therapies focused on the stimulation of cognitive functions, and work on adaptive skills. Since the intervention uses a smart, remote-controlled projector without a headset, the exposure to infection can be avoided following the COVID-19 pandemic-induced physical distancing policy in hospitals [13].

These experiences have proven to be very beneficial in various specialties in the field of mental health, but also in other areas such as pediatrics [14], neurology [15], palliative care and geriatric care [16,17]. Interestingly, recent studies investigated the use of VR for diminishing stress and anxiety among healthcare workers during the COVID-19 pandemic. VRelax, which involves 360° videos of calming natural environments watched via an HMD, effectively reduced stress and induced positive emotions in a sample of ICU nurses [18]. In addition, Tranquil Cinematic-VR simulation, which involves a 3-min 360° immersive video of a nature scene, lowered stress among frontline healthcare workers in COVID-19 treatment units [19]. Although the evidence is promising, there are still some limitations in the application of these techniques in health centers. The reluctance of professionals to participate in face-to-face sessions at the centers and the video content of the VR open the door to different approaches.

Anxiety-related disorders are the most common disorders in mental health. In the US, 18% of the population is considered to suffer from them [20]. According to the National Health Survey 2017, in Spain 6.7% of adults report suffering from chronic anxiety [21]. Conventional treatments for this type of disorder include medication, exercise, meditation and cognitive behavioral therapy. In recent decades, there has been an increasing interest in the study of burn-out syndrome in professionals and its consequences on mood. In this research, we focus on anxiety symptoms independent of burnout since the repercussions of anxiety on the quality of life and professional performance are well known. This study aimed to analyze the effect of immersive therapy through VR experience on the levels of anxiety of healthcare professionals working in a regional hospital in Olot (Spain). As secondary objectives, we wanted to analyze the effect of the intervention on healthcare professionals’ well-being (and its physical and emotional components) and the associations between clinical improvement and the presence of a history of psychological disorders, active psychological treatment, active pharmacological treatment and the number of treatment sessions.

## 2. Materials and Methods

### 2.1. Design and Participants

This was a quasi-experimental pilot study (with one intervention group and no control group) conducted in a regional hospital (Fundació Hospital d’Olot i Comarcal de la Garrotxa) in Spain. Professionals of the hospital as well as from the associated primary care centers who were active on 22 March 2021 were invited to participate in the study through an institutional email. Therefore, a non-probabilistic convenience sampling was used with the intention to reach at least 30 participants. Active professionals who accepted participation and signed the informed consent were included. The exclusion criteria of the study were limited to obtaining between 0 and 5 points on the Hamilton scale, i.e., people with no levels of anxiety. Among the 60 professionals who were interested in the project, seven did not show up at the start of the research appointment and 12 more did not meet the inclusion criteria. Therefore, 41 participants started the study with a drop-out rate of 14% at the end of the study (see Figure 1), due mainly to lack of time or motivation.

### 2.2. Ethical Aspects

The study was approved by the Clinical Research Ethics Committee (CEIm Girona) with code CEIM 2021.058. All participants gave written informed consent before data collection started. The participants could leave the research at any time with the sole commitment of notifying the psychologist.

### 2.3. Intervention

The intervention was a VR experience using a projection device with light and sound control that can provide an immersive experience covering the user’s field of vision (180° horizontal view × 120° vertical view) with full high-definition resolution. The software provides a customized interactive visual and auditory experience accessed by a smartphone application. Immersive therapy intervention was implemented through virtual reality for 8 weeks. In this first visit, the participants were informed about the intervention, which consisted of offering an immersive therapy appointment lasting approximately 10 min twice a week (recommended, extended at convenience and controlled). The hospital set aside an open space for wellness, with open access 24 h a day for professionals, where the virtual reality projector and a couple of armchairs were located to offer maximum comfort to the participants. The room, completely dark, allowed for complete immersion through the visual and auditory multisensory experience offered by the immersive therapy, creating an environment that enhances self-awareness to approach anxiety management. All intervention sessions took place in this wellness space. The participant autonomously controlled the management of the projections according to his or her needs, always within a selection of videos previously made by the research team.

### 2.4. Outcome Measurements

In a first appointment, the psychologist explained the project and the intervention, presented the informed consent and once the participant signed it, data was collected through a 20-min interview. Data collection was carried out by the center’s psychologist using a data collection notebook, complying with the data protection requirements, which were subsequently transferred to the corresponding spreadsheet. The registration of the number of sessions was carried out using a QR code that allowed each participant to anonymously register the number of sessions conducted on a weekly basis.

The dependent variable of the study was anxiety, evaluated with the Hamilton scale [22] validated in Spanish (in the public domain). The Spanish version of this instrument presents good psychometric properties, e.g., Cronbach’s alpha = 0.89, sensitivity to change = 1.36 and intraclass correlation coefficient = 0.92 [23]. The Hamilton quantitative scale was used to measure anxiety and administered by a psychologist through an interview. Three measurements were made, one before the start of the intervention, one at a 4-week follow-up and the final one at 8 weeks. The Hamilton scale consists of 14 items assessing anxiety at the somatic level (items 7, 8, 9, 10, 11, 12 and 13) and psychic level (items 1, 2, 3, 4, 5, 6 and 14). This scale considers three categories of anxiety: a value between 0 and 5 indicates no anxiety; between 6 and 15 indicates mild anxiety; and a value higher than 15 indicates moderate or severe anxiety [22].

Personal well-being was a secondary outcome, assessed by the Eudemon scale, which was applied pre- and post-intervention (at 8 weeks) [24]. Fierro and Rando assessed the psychometric properties of the Spanish version of the long 33-item scale and obtained Cronbach’s Alpha coefficients of 0.91 and 0.79 for the first and second components, respectively [25]. The item analysis showed that it can be reduced from 33 to 24 items without reducing the scale’s internal consistency (Cronbach’s Alpha coefficient = 0.92) or reliability [26]. Therefore, we used the 24-item version, with a 4-point Likert scale that assesses perceived subjective well-being.

Other variables tested were age, sex, psychological history, active psychological or pharmacological treatment and the number of virtual reality sessions carried out during the study. All the variables of this study were collected by the same center’s psychologist by consulting the information system or the participants’ self-reported.

### 2.5. Data Analysis

The Shapiro–Wilk and Kolmogorov–Smirnov normality tests were carried out with a significance result of less than 0.05 in almost all the variables, which indicated that non-parametric tests were more recommended. A descriptive analysis of the dependent and sociodemographic variables was carried out, indicating the mean, median, interquartile range and standard deviation. A bivariate analysis was also performed to compare the results at pre-, post-intervention and follow-up, using the non-parametric Wilcoxon test. The Hedge’s g statistic was used to measure the effect sizes for the difference of scores between the assessments. A clinical response was considered to exist when there was a positive change in the range of the Hamilton scale [22]. The analysis was performed with the IBM SPSS Statistics 21 software (IBM, Armonk, NY, USA). The confidence level was established at 95% (*p* < 0.05).

## 3. Results

As shown in Table 1, the final sample consisted of 35 professionals, all of them women with a mean age of 45.70 years. The professional profiles varied: 12 (34.3%) were nurses, 8 (22.8%) administrative assistants, 3 (8.6%) assistant nurses, 1 (2.9%) a physiotherapist, 3 (8.6%) physicians, 3 (8.6%) administrative staff (with higher education completed), 1 (2.9%) a social worker, 2 (5.6%) cleaning and security staff, 1 (2.9%) a psychologist, and 1 (2.9%) a radiological technician. Although the recommendation was to perform two weekly sessions, some heterogeneity was found in the number of sessions of each participant, which was not significantly associated with the decrease in anxiety or the improvement of well-being (*p* > 0.05).

Statistically significant differences (*p* < 0.001) were found in the decrease of anxiety after the application of the intervention, with a large effect size (g Hedges = 0.898); the median pre-intervention Hamilton score was 16.83 (range of 40), which was higher than the post-intervention score (median of 9.20; range of 28). When disaggregating the Hamilton scale into somatic and psychic symptoms, the statistical analysis indicated that the decrease in anxiety at the psychic level was significant (*p* < 0.001), with a significant effect size (g Hedges = 0.856); the median baseline score was 10.26 (range of 22) and the median score after the intervention 5.314 (range of 15). At the somatic level, the decrease was also significant (*p* = 0.001) and the effect size was moderate (g Hedges = 0.583), with median baseline and post-interventions scores of 6.57 (range of 21) and 3.91 (range of 17), respectively. Regarding the effect on well-being, a statistically significant (*p* < 0.001) increase of seven points on the Eudemon scale was found but with a moderate effect size (g Hedges = 0.629), in which the median baseline scores (median of 78.91 and range of 51) were lower than the post-intervention scores (median of 86.76 and range of 46).

After the intervention, 80.0% of participants had mild or no anxiety. Clinical changes in anxiety levels were significant (*p*-value of 0.002) according to the Chi-squared test (see Table 2).

Table 3 shows the association between clinical response and age, pharmacological and psychological treatment, history of psychological disorders and number of immersive therapy sessions undertaken. The associations were not statistically significant, but individuals with no history of psychological disorders nor current psychological treatment were more likely to have a positive clinical response. A complementary analysis was undertaken, using Spearman’s correlation between these independent variables (age, pharmacological and psychological treatment, history of psychological disorders and number of sessions) and the decrease in the Hamilton scale. Once again, associations were not significant, but those professionals with a history of psychological disorders or under current pharmacological or psychological treatment were less prone to have a positive clinical response.

## 4. Discussion

The main purpose of this study was to analyze the effects of immersive therapy with MK360 VR in the levels of anxiety and well-being of health professionals working in a regional hospital in Spain. Among the 35 professionals assessed, 85.7% improved their anxiety levels by decreasing their score on the Hamilton scale. This improvement in the score is accompanied by clinical relevance according to Hamilton’s criteria, moving from higher to lower anxiety ranges and a significant effect size. Furthermore, an increase in the score of the Eudemon scale was identified in approximately 83% of participants.

In the last decade, VR technology and its applications in the field of healthcare and particularly in psychology have increased rapidly. Different studies have indicated the effectiveness of VR in addressing psychological disorders and improving people’s well-being. One of the most prominent potentials of this technology is the exposure of the user to natural environments [27]. On this point, Attention Restoration Theory suggests that exposure to natural environments can improve stress and anxiety levels and improve work performance [28]. In this regard, Allison Anderson et al. concluded that VR therapy improves the level of relaxation both objectively and subjectively [29]. Other studies identify the potential of immersive VR as a novel strategy in psychological interventions [30,31,32]. The trend in recent years is to study the applications of this resource in cognitive disorders such as dementia [33] or Alzheimer’s disease [16]. In the case of health professionals, more recent studies suggest that VR is effective in reducing short-term stress [19,34].

To date, there are several studies specifically assessing VR to improve the state of anxiety in users. In this line, the systematic review conducted by Freeman et al. [35] about VR applications in mental health identified 154 intervention studies, which mostly analyzed anxiety. The studies conducted so far have demonstrated a slight superiority of VR versus conventional interventions for anxiety levels. However, no cost–benefit studies were identified in this review, and this type of research may be key in its implementation [36]. Another review, stricter in the inclusion criteria, was published in 2018 about VR exercises [37]. It included two randomized controlled trials, two quasi-experimental studies without a control group and one quasi-experimental study with a control group, four of which showed positive results. More recently, a systematic review has included some randomized clinical trials with larger samples [38].

The results of our study indicate that the two aspects of anxiety—psychic and somatic—improved significantly, and the difference between the values of both suggests that the improvement in the somatization of psychological processes (e.g., gastric problems, tremors) is lower. The existing literature has analyzed the effects of VR on anxiety without considering the symptomatology at the somatic and psychic level. So far, the treatment of somatic anxiety is mainly focused on symptom management and cognitive behavioral therapy, but it is difficult to manage [39,40]. In this regard, VR can be one more resource in the strategy of approaching both psychic and somatic anxiety, but more scientific evidence on this emerging therapy is necessary.

As a secondary objective, we wanted to analyze the association between clinical improvement and the presence of a history of psychological disorders, active psychological treatment, active pharmacological treatment and number of treatment sessions. No significant differences have been found in this regard; therefore, the decrease in the Hamilton score and the increases in well-being are generalized, even showing a tendency to greater effectiveness in volunteers without a history of psychological disorders and without active anxiety-related treatment. This finding suggests that the use of this novel strategy has a positive impact on unidentified and/or diagnosed anxiety (not necessarily mild anxiety) and as a broad-spectrum therapeutic resource to improve staff well-being. Although the research supports the results obtained in other studies conducted in recent years in the use of VR as a support in the treatment of anxiety, this study’s approach to undiagnosed or non-pathological anxiety provides new information not considered in most publications, which focus on pathological [41,42,43,44] or social anxiety [45,46,47].

Among the above-mentioned variables, the number of sessions has not shown to be a determining factor for the improvement of anxiety levels. This suggests that it is difficult to establish a general protocolized guideline for the application of this intervention, and this should be established by the participants themselves according to their needs and possibilities. Neither does the research conclude the suitability of the duration of the intervention since an improvement was shown both in the fourth and eighth week of intervention. Everything seems to indicate that there is a rapid improvement in anxiety and well-being, but more research is needed in this regard with longer follow-up periods.

Regarding limitations, the main one is the lack of a control group in this study, which was not considered feasible if we were to offer emotional support to all professionals willing to participate during an especially stressful (pandemic) period. In this regard, a Hawthorne effect and social desirability response bias cannot be ruled out, although effect sizes were moderate to large for the two main variables. Matching the experimental group with a control group receiving no intervention or usual care would have facilitated the isolation of the real effect of this novel intervention. Due to the small sample size, it was not possible https://www.youtube.com/c/DeadwingDuck (accessed on 3 June 2022) to undertake multivariate analysis. Future randomized controlled trials with larger sample size are warranted to further analyze the effects of immersive therapy.

## 5. Conclusions

In conclusion, immersive therapy using MK360 technology improves the participants’ experience, allowing them to relax and improve their emotional well-being and thus increase the sense of care and self-care of the organization’s professionals. The results obtained encourage us to carry out other research with this technology in the field of pain perception and its use in patients with pathologies related to cognitive functions and adaptive abilities.

## Figures and Tables

**Figure 1 ijerph-19-09793-f001:**
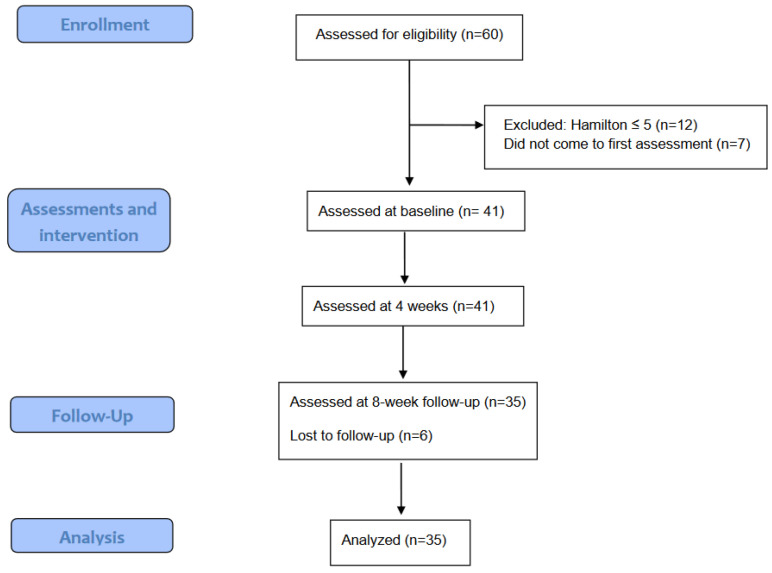
Flow chart of the sampling process.

**Table 1 ijerph-19-09793-t001:** Descriptive analysis of the sample (*n* = 35) of health professionals of a regional hospital in Spain.

Variable	Median	25th Percentile	75th Percentile	Interquartile Range
Age	47	40	51	11
H_1_	18	10	22	12
H_2_	12	8	15	7
H_3_	8	4	14	10
E_1_	82	70	90	20
E_3_	90	84	94	10
Number of sessions	10	8	14	6

H_1_ = score of Hamilton test at baseline; H_2_ = score of Hamilton test at 4-week follow-up; H_3_ = score of Hamilton test at 8-week follow-up; E_1_ = score of Eudemon test at baseline; E_3_ = score of Eudemon test at 8-week follow-up.

**Table 2 ijerph-19-09793-t002:** Clinical changes in anxiety levels and well-being between baseline (H_1_) and post-intervention (H_3_) assessments among professionals.

Anxiety Levels	H_1_ (*n*, %)	H_3_ (*n*, %)
No anxiety	0	12 (34.3)
Mild anxiety	17 (48.6)	16 (45.7)
Moderate-severe anxiety	18 (51.4)	7 (20.0)

**Table 3 ijerph-19-09793-t003:** Association between clinical response and age, pharmacological and psychological treatment, history of psychological disorders and number of immersive therapy sessions undertaken by professionals (Olot-Spain, 2021).

Independent Variables		*n* (%)	Positive Clinical Response (*n* = 20) *n* (%) or Median (Interquartile Range)	Non Clinical Response (*n* = 15) *n* (%) or Median (Interquartile Range)	*p* (U Mann-Whitney)	*p* (X^2^)
History of psychological disorders	Yes	17 (48.6)	8 (40.0)	9 (60.0)	-	0.241
	No	18 (51.4)	12 (60.0)	6 (40.0)		
Current psychological treatment	Yes	8 (22.9)	3 (15.0)	5 (33.3)	-	0.201
	No	27 (77.10)	17 (85.0)	10 (66.7)		
Current pharmacological treatment	Yes	13 (37.1)	7 (35.0)	6 (40.0)	-	0.762
	No	22 (62.9)	13 (65.0)	9 (60.0)		
Age		-	47 (15.5)	48 (10.0)	0.283	-
Number of sessions		-	10.0 (6.75)	11.0 (10.0)	0.254	-

## Data Availability

Not applicable.

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
