# Peer review of "Immersive Therapy for Improving Anxiety in Health Professionals of a Regional Hospital during the COVID-19 Pandemic: A Quasi-Experimental Pilot Study"

_ijerph, 2022, doi:10.3390/ijerph19169793_

Round 1
Reviewer 1 Report
Thank you for your clarifications. Now the results are more consistent and reliable.
Author Response
We thank the reviewer for all the comments to help us to improve the manuscript.
Reviewer 2 Report
The crucial issues (i.e., this is not an immersive application, and there is not a control group matched with the experimental group) have not been addressed. Correspondingly, I persist to my initial suggestion that this paper should be rejected.
Author Response
3. The critical issues of this study highlighted by Reviewer 2 need to be dealt with in more detail among the limits and suggestions for future research (i.e., this is not an immersive application, and there is not a control group matched with the experimental group). They are only partly quickly hinted at.
Answer:
As We explained in last revision, We think that the intervention can indeed be classified as Immersive Virtual Reality and We added new text to the manuscript, specifically in the Introduction and Methods (Intervention) section, as well as new citations. The title of the manuscript was also been modified.
The limit of not having a control group is highlighted in:
ï‚· Material and Methods: “This was a quasi-experimental pilot study (with one intervention group and no control group)…”.
ï‚· Discussion: “Regarding limitations, the main one is the lack of a control group in this study…”And: “Matching the experimental group with a control group receiving no intervention or usual care would have facilitated the isolation of the real effect of this novel intervention.”
Suggestions for future research are also raised in:
ï‚· Abstract: “Further studies with a control group are necessary to further analyze this novel intervention.”
ï‚· Discussion: “Future randomized controlled trials with larger sample size are warranted to further analyze the effects of immersive therapy.”
Reviewer 3 Report
Given the additions to the text, the references should be arranged in a certain order. Indications with place and year (tables 2 and 3) should be removed so as not to confuse readers.Author Response
. Given the additions to the text, the references should be arranged in a certain order (As suggested by
Reviewer 3).
Answer: We thank the reviewer for this suggestion. We have rectified the references’ order.
2. Indications with place and year (tables 2 and 3) should be removed so as not to confuse readers (As
suggested by Reviewer 3):
Answer: indications with place and year (Olot, 2021) have been removed in all tables.
Reviewer 4 Report
The authors have addressed the previous comments adequately.
Author Response
We thank the reviewer for all the comments to help us to improve the manuscript.
This manuscript is a resubmission of an earlier submission. The following is a list of the peer review reports and author responses from that submission.
Round 1
Reviewer 1 Report
The paper addresses important issues related to health of healthcare professionals during the COVID-19 pandemic. The therapy, suggested in the paper seems quite a promising approach. Still, I would like to point some comments/ questions.
- The background is really short and addresses only VR and some statistics on anxietym while no attention is given to the way researchers approach well-being. There are a lot of approaches to well-being but there is no rationalle in the paper, why you've chosen the approach you use.
- I believe the paper would benefit, if you would give a more detailed description of the intervention you used. What exactly they were experiencing in VR? Or any VR would have the effects you are describing?
- I have some concerns, if well-being can dramatically change in just 8 weeks? Could you give a more detailed arguments why you expected such a change?
- In the results section a lot of attention is given to anxiaety and disorders, but few to well-being. I would suggest either to broaden the part about the well-being, or maybe exclude it and concentrate on anxiety and disorders analysis.
- Finally, you associate the anxiety to COVID-19 pandemic, but since there is no data about the anxiety and well-being levels of those participants before the pandemic, how can you know that it was pandemic that increased this level? In my opinion, even if you take out the pandemic part (which is now very popular), the importance of the paper will remain since healthcare professional experience higher levels of stress compared to common people and their functionality affects everyone comming to the hospital.
Reviewer 2 Report
The article claims that this is an immersive virtual reality (IVR) system, and that there is scarcity of other IVR systems for reducing anxiety/stress. Both claims are false. Firstly, this is not an IVR system, it is just a projection of images/videos on the walls. Notably, based on the provided description, which is really vague and short, the system is not a Cave Automatic Virtual Environment (CAVE) system. In a CAVE system, the projection is also on the floor and the ceiling. Also, the CAVE systems require special VR glasses which make the objects 3D. Importantly, in a CAVE system, or any other IVR system (e.g., with VR HMDs) there is interaction with the virtual environment and its components. Finally, I am aware that there are several studies that have used proper IVR systems, with a decent sample size, and appropriate research design (e.g., with control groups). For these reasons, I do not see any merit in this study. I would suggest to the authors to collect data from a matched control group, and then to re-attempt to publish their work. However, in their new work, the authors should 1) avoid calling their system IVR (better to use a term like 3D projection system), 2) describe explicitly their system and the virtual environment that is projected 3) present and discuss all the relevant literature, and 4) consider and discuss issues such as cybersickness, novelty effect, and music/sounds.
Reviewer 3 Report
It is important research how to help to health care professionals. Thanks to the authors for it. I have some comments about this manuscript to improve it.
Introduction.
It is unclear whether immersive therapy by VR has been used in the past to assist healthcare professionals. The authors point to different specialties, but this applies to patients. The authors also report patient data for anxiety. Is there no data about professionals?
Materials and methods.
Participants – it should be shown how many participants were interviewed in total and how many were excluded according to the exclusion criteria.
Intervention – it is not clear what was done by stating that “data was collected through a 20-minute interview”. What data? What did the interview contain?
Outcome measurements – the authors describe the Hamilton scale, but do not describe the Eudemon scale, how many items it contains, whether it has subscales, and so on.
Results.
What the authors describe at the beginning of the chapter does not refer to the results but to the method and should be described under Materials and methods as a sample description.
It is not clear why the authors use the reference (Olot, 2021) to Table 1. What has it got to do with this study? The same applies to the titles of Tables 2 and 3.
It would be recommended to recalculate Table 3 - not vertically, but horizontally - so if there are 17 respondents with History of psychological disorders, then how many of them (n and %) are with Positive clinical response and how many with Non clinical response, etc.
Discussion.
The authors attribute the improvements to the intervention used, but more than half (21 out of 35) of the respondents used adjunctive therapy, and this variable was not controlled.
Self-reported surveys, which allow for socially desirable responses, should also be definitely mentioned as a limitation of the study.
Reviewer 4 Report
Thank you for the opportunity to review this interesting manuscript. I have a few comments for the authors to consider:
· Abstract: lines 23-26: Please state the actual numbers when reporting the results, for example, what was the effect sizes? 0.5? 0.8?
· Introduction:
o It would be good for the authors to provide a stronger rationale to studying the health professionals in Spain when the cited issue (E.g. prevalence of anxiety-related disorders) was based on the US data (lines 77-83). Please provide the prevalence in Spain, clarify the extent of the conventional treatments used in Spain, the extent of burnout in Spain, and note if there is a research gap in using VR in Spain (or in Olot) to highlight the need to do this study.
o Line 86: Please change the word “correlation” to “associations”.
· Methods:
o Please clarify if there is a control group, and if they received usual care or nothing at all.
o Please add the validity or reliability score of the tools used for all the outcome measures (131-142).
o Regarding the other variables listed from lines 143-144, please clarify how the research team collect data around active psychological or pharmacological treatment. Was this from contacting the participants’ doctors, or self-reported by the participants? The research team may have to note misclassification bias or social desirability bias as a potential bias of the finding under the Discussion/Limitations section.
o The authors could have transformed the skewed variables and perform a repeated measures analysis or apply clinical-sensible cut-off points to the outcome variables and perform logistic regression whilst adjusting for the effects of other covariates. Please provide a justification for not performing the statistical analyses as suggested, whilst adding a limitation that this study could have issues with confounding bias, or update the analyses (and results) accordingly,
o The authors have yet included the description of the statistical analyses performed to assess their secondary objective. Please add details. For example, association between clinical response (positive/none) and history of psychological disorders (yes/no) was assessed using a Chi-squared test (or Fisher’s Exact test). Similarly, the associations between clinical response and active psychological treatment, active pharmacological treatment and number of treatment sessions were performed using separate Chi-squared tests (or Fisher’s Exact tests). Furthermore, the overall associations between VR and clinical response were performed using logistic regression models, adjusted for presence of a history of psychological disorders, active psychological treatment, active pharmacological treatment and number of treatment sessions to reduce potential confounding bias.
· Results:
o Table 1: Please note that mean and standard deviation, as well as 95% CI are not recommended for skewed variables. Do remove the columns from Table 1. The authors can just report the median, and the range from the 25th to the 75th percentile so that the readers can gather some information across the percentiles, rather than a subtracted number of interquartile range.
o Line 198: Please remove “linear-by-linear”.
o Table 3: Please add a footnote to clarify the statistical tests performed, e.g. Logistic regression, and include the adjusted odds ratio.
· Discussion:
o Lines 251-259: Please revise this paragraph so that the discussion is about the updated results. Please also focus on discussing the results in relation to other literature, instead of repeating the results.